# A Bayesian Approach to Quantify the Uncertainty of Human Ratings in a Single-Instance Multimodal Framework

**Zijian Chen** [1]  **Archana Venkataraman** [1]

## Abstract

Human ratings are central to learning and inference across several application domains, but they are also subject to inter-rater biases and judgment errors. Quantifying the uncertainty of these human ratings would require repeated measurements, which are expensive and rarely available at scale. We propose a Bayesian graphical model to estimate the instance-level and item-level uncertainty of (subjective) human ratings by leveraging auxiliary (objective) data. Our model learns a shared latent content representation that explains factors common to both the human rating and auxiliary data and a latent uncertainty variable that captures fluctuations in the human assessments via a data-conditioned prior. We develop a scalable amortized variational inference procedure that uses modality-appropriate neural encoders and decoders to represent the posterior factors. Experiments on synthetic data demonstrate that our framework can accurately recover the latent uncertainty under targeted ablations and stress tests. We further demonstrate our approach on a real-world dataset of paired functional MRI scans and behavioral testing for autism, thus highlighting the need for uncertainty quantification.

## 1. Introduction

Human ratings play a central role in decision-making pipelines across many application domains. For example, a clinician might rate the severity of an illness, which directly shapes downstream diagnosis and treatment plans; legal review of regulatory compliance will impact the design and launches of new products; human rating quality has a large effect on supervised AI model training (Born-

---

[1]Department of Electrical and Computer Engineering, Boston University, Boston, Massachusetts, USA. Correspondence to: Zijian Chen <zijianc@bu.edu>.

*Proceedings of the 43rd International Conference on Machine Learning*, Seoul, South Korea. PMLR 306, 2026. Copyright 2026 by the author(s).

mann et al., 2010; Walton et al., 2015). Despite their importance, human ratings are inherently subjective and can vary across people, contexts, and time. Moreover, such variability is *item-specific*: some items admit stable assessments, while others are ambiguous or sensitive to small contextual changes (Uma et al., 2021). Thus, treating human ratings as reliable can distort confidence estimates and lead to questionable downstream decisions. Conversely, if we can quantify the rating uncertainty at the item level, this information can be used to re-calibrate downstream tasks.

One way to quantify the uncertainty of a given human rating is to collect repeated measurements from different raters (Shrout & Fleiss, 1979; De Vet et al., 2011). However, this approach can be expensive and impractical at scale (Sheng et al., 2008). An alternative strategy is to leverage *auxiliary (objective) data* collected alongside the human rating (Jin et al., 2017). Such data can be high-dimensional (e.g., imaging, documents) and often contains information about the underlying item state. While this data may be insufficient to replace the human ratings, they can be used to infer the stability of a given rating item (Liu et al., 2021).

One illustrative example is precision psychiatry, where clinically-relevant variables, such as symptom severity, are derived from structured behavioral assessments administered and interpreted by trained psychologists. These assessments be influenced by inter-rater differences, test-retest effects, and patient cooperation (Faherty et al., 2020). One source of auxiliary data in this field is neuroimaging (e.g., MRI, fMRI), which provides a rich objective signal of changes in the brain that can be compared with the behavioral assessments to determine reliability (Prescott, 2013).

From a technical standpoint, there are two coupled challenges that we must address. The first challenge is to quantify the item-level uncertainty of the subjective ratings *without requiring multiple human measurements*. In the precision psychiatry example, some behavioral traits are easier to query than others, which leads to varying uncertainty. However, psychologists are expensive, so it is unlikely a patient can be evaluated multiple times. The second challenge is to determine which features of the objective data are relevant to uncertainty estimation. Specifically, auxiliary data, like neuroimaging, can be high-dimensional, heterogeneous, and

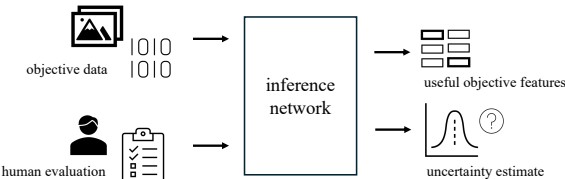

*Figure 1.* High-level overview of our approach. Our framework takes as input paired human ratings and auxiliary (objective) data. It outputs the item-level uncertainty estimates for the human rating and identifies the informative objective features.

dominated by non-informative or degraded components. If the estimation procedure absorbs this irrelevant information into its representation, then the inferred uncertainty can erroneously equate instability in the objective observation with uncertainty in the human rating process.

### 1.1. Our Contributions

We propose a novel Bayesian framework for paired human ratings and auxiliary data that simultaneously estimates the item-level uncertainty of human ratings and identifies relevant auxiliary features that guide the estimation (Fig. 1).

Our framework constructs a shared *latent content representation* that captures the common factors of both the subjective ratings and the auxiliary objective evidence, such as imaging data, while introducing two complementary latent variables that play distinct roles. First, an item-specific uncertainty modulates the distribution of human ratings and is designed to capture the variation implied by the posterior predictive distribution of repeated ratings. Second, a learnable mask acts as a feature selector over the objective data by gating the components that are explained through the shared representation versus those explained by a noise mechanism independent of the content. Together, these choices produce robust uncertainty estimates for the human ratings, while learning a meaningful feature set in the objective data.

We develop a scalable amortized variational inference algorithm that uses deep networks to implicitly learn the posterior of the latent content, uncertainty, and feature-selection masks. Amortization replaces per-instance variational parameters with encoder networks that map each paired observation to an approximate posterior distribution over the latent variables (Margossian & Blei, 2024). This approach supports large datasets, enables flexible conditioning on available modalities, and preserves a coherent probabilistic interpretation through a variational evidence objective.

We validate our framework through both simulation and real-world studies. On synthetic data with known ground-truth information, we show that our framework recovers the intended latent variable decomposition. Targeted ablations and stress tests demonstrate the value of each model com-

ponent. On paired neuroimaging and behavioral data for autism, our framework improves the conditional prediction of human ratings (i.e., empirical predictive intervals coverage and point estimation), as compared to other uncertainty-aware baselines including heteroscedastic regression, deep ensembles, evidential regression, and multimodal models.

## 2. Related Works

### 2.1. Aleatoric Uncertainty Estimation

Aleatoric uncertainty describes the irreducible variability inherent to the data. One approach to mitigate aleatoric uncertainty in the context of supervised learning is *heteroscedastic regression*. Here, a model is trained to output an input-conditioned variance, which rescales the loss terms associated with noisy (i.e., high variance) samples (Kendall & Gal, 2017; Young et al., 2025). Building off this premise, the work of (Collier et al., 2021) introduces a latent structure at the output, so that the model can capture correlated, input-dependent variability. Our approach borrows the input-conditioned noise principle but explicitly encodes the uncertainty in data generation rather than simply predicting a noise variance. An alternative to heteroscedastic regression is to train an evidentiary model to outputs either the parameters of an uncertainty distribution (Amini et al., 2020) or the voxel-level aleatoric uncertainty; this information is used to improve model training (Shi et al., 2024). Our framework also predicts distributional parameters but integrates them into data modeling rather than a final endpoint.

Beyond supervised tasks, representation learning has also been adapted to model a distribution over the latent factors. This information allows downstream tasks to inherit a faithful notion of data ambiguity (Kirchhof et al., 2023). However, these distributional parameters cannot distinguish useful content from rating instability.

### 2.2. Multimodal Representation Learning

A second line of related work is to learn a joint model over the input data (e.g., auxiliary imaging) and the human labels (e.g., behavioral ratings) as an implicit regularization. One strategy, used in earlier works, is to learn a joint distribution over the input data and labels on the training set and use variational inference to generate labels for the unlabeled test samples (Kingma et al., 2014). Follow-up work strengthens this paradigm by incorporating discriminative criteria into the inference objective, while preserving the ability to sample in the joint space (Maaløe et al., 2016; Li et al., 2017). While seminal, these frameworks treat the labels as a ground-truth predictive target during training but do not model potential unreliability in the labels, themselves.

More broadly, the joint modeling of input data and human labels allows us to couple these information streams through

a shared latent structure. For example, joint VAEs combine neural decoders for images with Bayesian SVM modules for the supervision signal to support cross-modal inference (Pu et al., 2016). Recent extensions of VAEs include multi-task diffusion models that share a denoising backbone to capture the joint distribution over images and labels (Chen et al., 2024). Meanwhile, energy-based models have been revisited as a joint modeling approach that reinterprets the classifier logits as an energy function over image-label pairs for joint density learning (Yang et al., 2023). These generative approaches offer a powerful way to model high-dimensional structured data. However, the black-box nature of the neural encoders can weaken interpretability.

### 2.3. Label Uncertainty via Crowd-sourced Data

Finally, we can directly quantify the uncertainty of human-derived information by comparing multiple evaluations. Classic models treat each annotation as a noisy observation and estimate the annotator-specific reliability using standard Bayesian inference techniques (Dawid & Skene, 1979), or output confusion matrices of annotator variability (Rodrigues & Pereira, 2018; Ibrahim et al., 2023; Guo et al., 2023). While arguably more accurate than joint modeling or representation learning, these methods require several repeated measurements. This data can be impractical to collect, as described in the precision psychiatry example.

## 3. Methods

At a high level, our framework treats each human rating as the result of two factors: the underlying content of the item being evaluated, and the uncertainty introduced by the human assessment process. In the case where repeated ratings are not available, the auxiliary objective data serves to provide evidence about the underlying content.

Table 1 summarizes the random variables and key parameters of our Bayesian framework. We use a standard convention, in which capital letters correspond to random variables, and lower-case letters denote a particular realization. Bold font symbolizes a either a matrix or vector, whereas non-bold font denotes a scalar. Greek letters are used to represent the parameters of a given probability distribution.

### 3.1. Generative Model

Fig. 2 illustrates our Bayesian graphical model. Formally, $\mathbf{X}_n \in \mathbb{R}^{d_X}$ denotes the vector of human ratings for sample $n$, and $\mathbf{Y}_n \in \mathbb{R}^{d_Y}$ represents the corresponding auxiliary objective data. The goal of our framework is to estimate the item-level uncertainty of the human ratings, which we formalize mathematically, as the posterior distribution over the latent uncertainty variable $\mathbf{U}_n$ given the observed variables, i.e., $p(\mathbf{u}_n|\mathbf{x}_n, \mathbf{y}_n)$. As a secondary goal, our framework

*Table 1.* Variables and parameters in our Bayesian framework.

| Notation | Description |
| --- | --- |
| $C$ | Subgroup Indicator |
| $\mathbf{X}$ | (Subjective) Human Rating |
| $\mathbf{Y}$ | (Objective) Auxiliary Data |
| $\mathbf{Z}$ | Latent Content Vector |
| $\mathbf{U}$ | Item-Specific Rating Uncertainty |
| $\mathbf{M}$ | Feature Selection Mask |
| $\omega, \theta, \phi, \psi$ | Distributional Parameters |

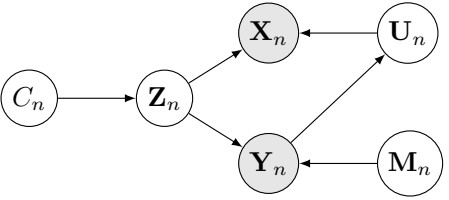

$$\mathbf{Z}_n \mid \{C_n = k\} \sim \mathcal{N}\left(\boldsymbol{\mu}_k, \boldsymbol{\sigma}_k^2 \mathbf{I}\right)$$
$$\mathbf{Y}_n \mid \mathbf{Z}_n, \mathbf{M}_n \sim p_\omega(\mathbf{y}_n \mid \mathbf{z}_n, \mathbf{m}_n),$$
$$\mathbf{X}_n \mid \mathbf{Z}_n, \mathbf{U}_n \sim p_\phi(\mathbf{x}_n \mid \mathbf{z}_n, \mathbf{u}_n),$$
$$\mathbf{U}_n \mid \mathbf{Y}_n \sim \mathcal{N}(\mathbf{0}, \Sigma_\psi(\mathbf{y}_n)),$$
$$\mathbf{M}_{n,i} \sim \text{Ber}(\rho_i), \quad i = 1, \dots, d_Y$$

*Figure 2.* The proposed directed graphical model to separate predictive content and rating uncertainty. Arrows indicate conditional dependencies, and shaded nodes are observed random variables.

also learns the features of $\mathbf{Y}_n$ that are relevant for the uncertainty estimation. This is feature selection is done via a latent mask $\mathbf{M}_n$, with desired posterior $p(\mathbf{m}_n|\mathbf{x}_n, \mathbf{y}_n)$.

Given the observed variables $(\mathbf{X}_n, \mathbf{Y}_n)$, the central modeling assumption is that the human ratings and auxiliary data are driven by a shared latent content variable $\mathbf{Z}_n \in \mathbb{R}^{d_Z}$ shown in Fig. 2. We place a mixture of Gaussians prior on $\mathbf{Z}_n$ to capture heterogeneous or state-dependent characteristics of the dataset. The number of mixture components can be set to one if this flexibility is not needed, thus downgrading the prior on $\mathbf{Z}_n$ to a standard Gaussian distribution.

The observed variables are generated based on $\mathbf{Z}_n$. In practice, the auxiliary data $\mathbf{Y}_n$ can be high dimensional (e.g., neuroimaging) and include components that are unrelated to the human ratings. We encode this selective association through a binary feature selection mask $\mathbf{M}_n \in \mathbb{R}^{d_Y}$. Specifically, $\mathbf{M}_{n,i} = 1$ indicates that feature $\mathbf{Y}_{n,i}$ is associated with a background/noise mechanism independent of $\mathbf{Z}_n$, and $\mathbf{M}_{n,i} = 0$ indicates that $\mathbf{Y}_{n,i}$ is explained by the latent content. We assume a Bernoulli prior on $\mathbf{M}_{n,i}$ and conditional

independence of each feature $\mathbf{Y}_{n,i}$ which yields:

$$
\begin{aligned}
&p_\omega(\mathbf{y}_n|\mathbf{z}_n, \mathbf{m}_n) \\
&= \prod_{i=1}^{d_Y} p_{\text{base}}(\mathbf{y}_{n,i})^{\mathbf{m}_{n,i}} p_{\text{cont}}(\mathbf{y}_{n,i}|\mathbf{z}_n)^{1-\mathbf{m}_{n,i}},
\end{aligned} \quad (1)
$$

where $p_{\text{base}}(\cdot)$ and $p_{\text{cont}}(\cdot)$ are the background and content-dependent distributions of the auxiliary data, respectively.

For the subjective data, we introduce an uncertainty variable $\mathbf{U}_n \in \mathbb{R}^{d_X}$ that captures the fluctuation of each human rating item beyond what is described by the latent content vector $\mathbf{Z}_n$. We assume an additive model for the $\mathbf{X}_n$ conditioned on the latent content and uncertainty as follows:

$$
p_\phi(\mathbf{x}_n \mid \mathbf{z}_n, \mathbf{u}_n) = \mathcal{N}\big(\mathbf{x}_n; \boldsymbol{A}\mathbf{z}_n + \mathbf{u}_n, \text{diag}(\boldsymbol{\sigma}_X^2)\big), \quad (2)
$$

where $\boldsymbol{A}$ is a learnable loading matrix. The linear-Gaussian form in Eq. (2) yields an interpretable decomposition: $\boldsymbol{A}\mathbf{z}_n$ captures the predictable (content-driven) component, while $\mathbf{u}_n$ captures item-specific uncertainty.

As a means of regularization, we ground $\mathbf{U}_n$ in the auxiliary data via an objective-conditioned prior:

$$
p_\psi(\mathbf{u}_n \mid \mathbf{y}_n) = \mathcal{N}\big(\mathbf{0}, \Sigma_\psi(\mathbf{y}_n)\big), \quad (3)
$$

where $\Sigma_\psi(\mathbf{y})$ is parameterized by a neural network with positive outputs (e.g., softplus on diagonal variances). We assume zero mean to prevent $\mathbf{U}_n$ from absorbing systematic contents that should be contained in $\boldsymbol{A}\mathbf{z}_n$ in Eq. (2).

### 3.2. Inference Procedure

For notational simplicity, we will drop the superscript $n$ when deriving the amortized variational inference algorithm and re-introduce it in our final objective function. The main goal of the inference procedure is to estimate the posterior probability distributions $p(\mathbf{u}|\mathbf{x}, \mathbf{y})$ and $p(\mathbf{m}|\mathbf{x}, \mathbf{y})$.

As shown in Fig. 2, the subgroup assignment $C$ is conditionally independent of $\{\mathbf{X}, \mathbf{Y}, \mathbf{U}, \mathbf{M}\}$ given latent content vector $\mathbf{Z}$. Thus, the posterior distribution of the latent variables can be factorized as follows:

$$
p(\mathbf{z}, \mathbf{u}, \mathbf{m}, c|\mathbf{x}, \mathbf{y}) = p(c|\mathbf{z})p(\mathbf{z}, \mathbf{u}, \mathbf{m}|\mathbf{x}, \mathbf{y}) \quad (4)
$$

such that

$$
p(c = k|\mathbf{z}) = \frac{\pi_k\, \mathcal{N}(\mathbf{z}; \boldsymbol{\mu}_k, \sigma_k^2\mathbf{I})}{\sum_{\ell=1}^{K} \pi_\ell\, \mathcal{N}(\mathbf{z}; \boldsymbol{\mu}_\ell, \sigma_\ell^2\mathbf{I})}. \quad (5)
$$

The parameters $\{\pi_k, \mu_k, \sigma_k^2\}$ can be updated according to the standard EM algorithm for Gaussian mixture models.

The second term of Eq. (4) is intractable, so we use a variational inference strategy with mean-field factorization:

$$
q(\mathbf{z}, \mathbf{u}, \mathbf{m}|\mathbf{x}, \mathbf{y}) = q(\mathbf{z}|\mathbf{x}, \mathbf{y})q(\mathbf{u}|\mathbf{x}, \mathbf{y})q(\mathbf{m}|\mathbf{y}, \mathbf{z}) \quad (6)
$$

---

**Algorithm 1** Amortized variational inference

**Require:** Paired data $\mathcal{D} = \{(\mathbf{x}_n, \mathbf{y}_n)\}_{n=1}^N$; mixture prior $\{(\pi_k, \boldsymbol{\mu}_k, \boldsymbol{\sigma}_k^2)\}_{k=1}^K$; mask prior $\boldsymbol{\rho}$; MC samples $L$
1: **for** minibatch $\mathcal{B} \subset \mathcal{D}$ **do**
2:     Encode $\mathcal{B}$ to get $h_{\cdot,\theta}(\mathbf{x}, \mathbf{y})$ and $s_{\cdot,\theta}(\mathbf{x}, \mathbf{y})$
3:     $q(\mathbf{z}|\mathbf{x}, \mathbf{y}),\ q(\mathbf{u}|\mathbf{x}, \mathbf{y}) \leftarrow h_{\cdot,\theta}(\mathbf{x}, \mathbf{y}), s_{\cdot,\theta}(\mathbf{x}, \mathbf{y})$
4:     Sample $\mathbf{z}^{(1:L)} \sim q(\mathbf{z} \mid \mathbf{x}, \mathbf{y})$ via reparameterization.
5:     Compute selection probability $\mathbf{g}^{(\ell)} \leftarrow g_\theta(\mathbf{y}, \mathbf{z}^{(\ell)})$ for each $\mathbf{z}^{(\ell)}$
6:     **Compute ELBO Terms**
    $\mathcal{L}_Y$ by MC samples $\{\mathbf{z}^{(\ell)}\}$ and $\{\mathbf{g}^{(\ell)}\}$;
    $\mathcal{L}_X$ analytically
    $\mathcal{D}_{\text{KL}}(q(\mathbf{z}|\mathbf{x}, \mathbf{y})\|p(\mathbf{z}))$ by MC samples $\{\mathbf{z}^{(\ell)}\}$;
    $\mathcal{D}_{\text{KL}}(q(\mathbf{u}|\mathbf{x}, \mathbf{y})\|p(\mathbf{u}|\mathbf{y}))$ analytically;
    $\mathbb{E}\mathcal{D}_{\text{KL}}(q(\mathbf{m}|\mathbf{y}, \mathbf{z})\|p(\mathbf{m}))$ by MC samples $\{\mathbf{g}^{(\ell)}\}$.
7:     Compute mask regularizers from $\mathbf{g}^{(\ell)}$
8:     Sum to obtain $\mathcal{L}_t$ and descend $-\nabla\mathcal{L}_t$ with AdamW.
9: **end for**
10: Update $(\pi_k, \boldsymbol{\mu}_k, \boldsymbol{\sigma}_k^2)_{k=1}^K$ via $q(\mathbf{z}|\mathbf{x}, \mathbf{y})$ (responsibilities from $p(c \mid \mathbf{z})$, then moment matching).

---

Given the forward model in Fig. 2, we assume Gaussian distributions for the approximate posteriors over $\mathbf{Z}$ and $\mathbf{U}$:

$$
q(\mathbf{z}|\mathbf{x}, \mathbf{y}) = \mathcal{N}(h_{z,\theta}(\mathbf{x}, \mathbf{y}), \text{diag}(s_{z,\theta}(\mathbf{x}, \mathbf{y})^2)), \quad (7)
$$
$$
q(\mathbf{u}|\mathbf{x}, \mathbf{y}) = \mathcal{N}(h_{u,\theta}(\mathbf{x}, \mathbf{y}), \text{diag}(s_{u,\theta}(\mathbf{x}, \mathbf{y})^2)). \quad (8)
$$

The mean functions $h_{z,\theta}(\cdot, \cdot)$, $h_{u,\theta}(\cdot, \cdot)$ and the standard deviation functions $s_{z,\theta}(\cdot, \cdot)$, $s_{u,\theta}(\cdot, \cdot)$ are implemented as neural networks. These networks encode the observed human rating $\mathbf{x}$ and auxiliary data $\mathbf{y}$ using modality-appropriate feature extractors, fuse the resulting representations, and output the respective Gaussian parameters.

The approximate posterior $q(\mathbf{m}|\mathbf{y}, \mathbf{z})$ for the feature mask $\mathbf{M}$ is modeled as a factorized Bernoulli distribution:

$$
q(\mathbf{m}|\mathbf{y}, \mathbf{z}) = \prod_{i=1}^{d_Y} g_\theta(\mathbf{y}, \mathbf{z})_i^{\mathbf{m}_i}(1 - g_\theta(\mathbf{y}, \mathbf{z})_i)^{1-\mathbf{m}_i}, \quad (9)
$$

where $g_\theta(\mathbf{y}, \mathbf{z}) \in (0, 1)^{d_Y}$ is implemented as a neural network that outputs the probability of each component being assigned to the background mechanism, which is considered irrelevant to the uncertainty estimation.

The evidence lower bound (ELBO) objective function given the observation $(\mathbf{x}, \mathbf{y})$ can be expressed as follows:

$$
\begin{aligned}
\mathcal{L} &= \mathbb{E}_q[\log p(\mathbf{x}, \mathbf{y}, \mathbf{z}, \mathbf{u}, \mathbf{m}) - \log q(\mathbf{z}, \mathbf{u}, \mathbf{m}|\mathbf{x}, \mathbf{y})] \\
&= \underbrace{\mathbb{E}_{q_Z}\mathbb{E}_{q_M}[\log p_\omega(\mathbf{y}|\mathbf{z}, \mathbf{m})]}_{\mathcal{L}_\mathcal{Y}} + \underbrace{\mathbb{E}_{q_Z}\mathbb{E}_{q_U}[\log p_\phi(\mathbf{x}|\mathbf{z}, \mathbf{u})]}_{\mathcal{L}_X} \\
&\quad - \mathcal{D}_{\text{KL}}(q(\mathbf{z}|\mathbf{x}, \mathbf{y})\|p(\mathbf{z})) - \mathcal{D}_{\text{KL}}(q(\mathbf{u}|\mathbf{x}, \mathbf{y})\|p(\mathbf{u}|\mathbf{y})) \\
&\quad - \mathbb{E}_{q_Z}\mathcal{D}_{\text{KL}}(q(\mathbf{m}|\mathbf{y}, \mathbf{z})\|p(\mathbf{m})). \quad (10)
\end{aligned}
$$

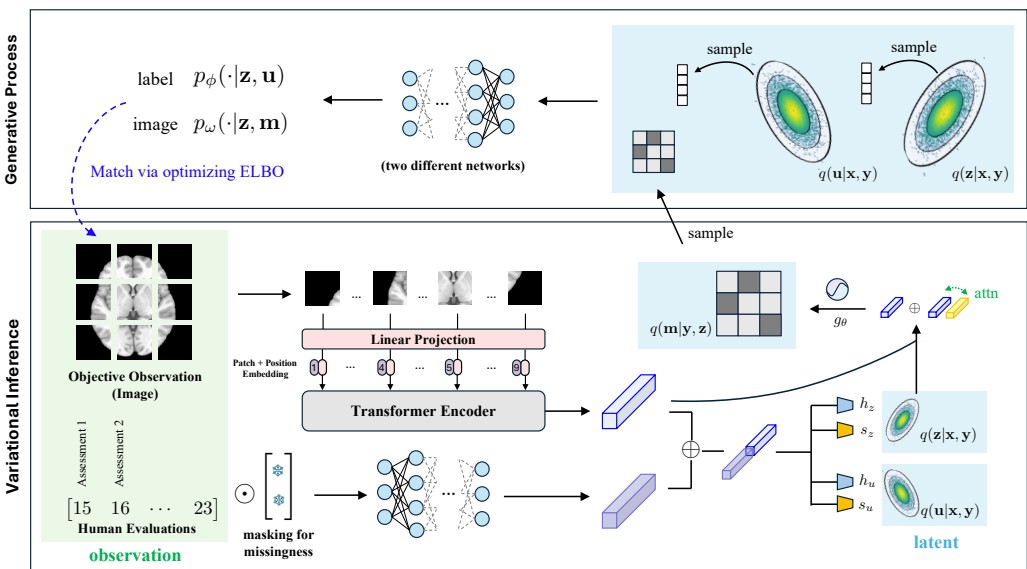

*Figure 3.* Model architecture for the inference networks. The auxiliary data in our experiments is assumed to be neuroimaging. However, our framework is agnostic to the encoding architecture, which can be re-designed for other data modalities.

Since we have a closed-form expression for $p(c|\mathbf{z})$ and since $\mathbf{Z}$ separates $C$ from the other random variables, we have marginalized the dependencies on $C$ in Eq. (10). Detailed expression for each of these terms are found in the appendix.

We denote the first two terms of Eq. (10) as $\mathcal{L}_Y$ and $\mathcal{L}_X$, respectively. The term $\mathcal{L}_Y$ is responsible for updating the background and content distributions $p_{\text{base}}(\cdot)$ and $p_{\text{cont}}(\cdot)$ of the conditional distribution $p_\omega(\mathbf{y}_n|\mathbf{z}_n, \mathbf{m}_n)$. Meanwhile, since $q(\mathbf{m} \mid \mathbf{y}, \mathbf{z})$ is parameterized by $g_\theta(\mathbf{y}, \mathbf{z})$, $\mathcal{L}_Y$ also updates the mask posterior via gradient flow into $g_\theta(\cdot, \cdot)$. The variational factor $q(\mathbf{z}|\mathbf{x}, \mathbf{y})$ for the latent content vector is also updated since $p_{\text{cont}}(\cdot)$ and $g_\theta(\cdot, \cdot)$ depend on $\mathbf{z}$. The term $\mathcal{L}_X$ in Eq. (10) updates the prior distribution $p_\phi(\mathbf{x}|\mathbf{z}, \mathbf{u})$ and thus $q(\mathbf{z}|\mathbf{x}, \mathbf{y})$ and $q(\mathbf{u}|\mathbf{x}, \mathbf{y})$.

The remaining terms in Eq. (10) explicitly regularize each posterior factor toward its corresponding prior. The KL-divergence on $\mathbf{Z}$ penalizes deviation of the inferred content distribution from the mixture of Gaussians structure. The KL-divergence on $\mathbf{U}$ encourages the mean to be zero and the variance to be explainable through $\mathbf{Y}$, while updating $p_\psi(\mathbf{u}_n|\mathbf{y}_n)$. Finally, the KL-divergence on $\mathbf{M}$ penalizes deviations from the Bernoulli sparsity prior, preventing degenerate solutions in which the mask arbitrarily flips to improve $\mathcal{L}_Y$ without paying a complexity cost.

Finally, we add an explicit regularization on the selection probabilities $g_\theta(\mathbf{y}, \mathbf{z})$ that controls the average fraction of components assigned to background/noise:

$$r(\mathbf{y}, \mathbf{z}) = \frac{1}{d_Y} \sum_{i=1}^{d_Y} g_\theta(\mathbf{y}, \mathbf{z})_i. \quad (11)$$

This term prevents degenerate solutions, in which all components are treated as informative, and it encourages a compact set of informative components. Combining the ELBO and the regularizers yields the overall training objective:

$$\mathcal{L}_t = \sum_{n=1}^{N} \mathcal{L}(\mathbf{x}_n, \mathbf{y}_n) - \lambda_s \sum_{n=1}^{N} \mathbb{E}_{q(\mathbf{z}|\mathbf{x}, \mathbf{y})} \left[ r(\mathbf{y}_n, \mathbf{z}_n) \right] \quad (12)$$

We maximize $\mathcal{L}_t$ as per variational inference. Algorithm 1 summarizes the computation performed for each epoch.

### 3.3. Implementation

The auxiliary data $\mathbf{Y}_n$ in our experiments corresponds to a vector of (flattened) neuroimaging data. Therefore, we assume Gaussian background and content distributions:

$$p_{\text{base}}(\mathbf{y}_n) = \mathcal{N}(\mathbf{y}_n; \boldsymbol{\mu}_{\text{base}}, \text{diag}(\boldsymbol{\sigma}_{\text{base}}^2))$$
$$p_{\text{cont}}(\mathbf{y}_n|\mathbf{z}_n) = \mathcal{N}(\mathbf{y}_n; \boldsymbol{\mu}_{\text{cont}}(\mathbf{z}_n), \text{diag}(\boldsymbol{\sigma}_{\text{base}}^2(\mathbf{z}_n))),$$

where the $\{\boldsymbol{\mu}_{\text{base}}, \boldsymbol{\sigma}_{\text{base}}^2\}$ capture content-independent variability, while $\{\boldsymbol{\mu}_{\text{cont}}(\mathbf{z}_n), \boldsymbol{\sigma}_{\text{cont}}^2(\mathbf{z}_n)\}$ are learned implicitly via multi-layer perceptrons (MLPs) that operate on the latent content $\mathbf{z}_n$. Importantly, our framework is agnostic to the choice of network used to learn the content distribution; these networks can be adapted based on the data in hand.

For the inference networks $q(\mathbf{z}|\mathbf{x}, \mathbf{y})$ and $q(\mathbf{u}|\mathbf{x}, \mathbf{y})$, we use separate encoders for the imaging data and human ratings. The imaging encoder is a Vision Transformer (ViT) and the rating encoder is an MLP. The fused representation of imaging and label is passed to two heads that produce $(h_{z,\theta}, s_{z,\theta})$ and $(h_{u,\theta}, s_{u,\theta})$. The prior $\Sigma_\psi(\mathbf{y})$ is generated

*Table 2.* Synthetic ablation and sensitivity study. We report mean±std over 10 random seeds on the validation split. Hard-$\mathbf{M}$ increases the corruption. Hard-$\mathbf{U}$ increases label ambiguity. Hard-$\mathbf{Z}$ makes the latent clusters less separable.

| Setting | RMSE$_X$ ↓ | RMSE$_Y$ ↓ | $R_Z^2$ ↑ | Acc$_C$ ↑ | $\rho(\hat{U}, U)$ ↑ | AUC$_M$ ↑ | IoU$_M$ ↑ |
|---|---|---|---|---|---|---|---|
| **Ablations (standard generator)** | | | | | | | |
| Full | **0.694±0.087** | **0.455±0.034** | **0.925±0.022** | **0.999±0.002** | **0.666±0.048** | **0.840±0.011** | **0.350±0.040** |
| $-\mathbf{U}$ | 0.853±0.084 | 0.462±0.037 | 0.914±0.025 | 0.997±0.001 | – | 0.836±0.013 | 0.342±0.041 |
| $-\mathbf{M}$ | 0.714±0.093 | 0.716±0.006 | 0.894±0.024 | 0.987±0.003 | 0.648±0.053 | – | – |
| $-\mathbf{U}, -\mathbf{M}$ | 0.818±0.101 | 0.715±0.009 | 0.914±0.010 | 0.931±0.150 | – | – | – |
| **Sensitivity (Full model; harder generator)** | | | | | | | |
| Hard-$\mathbf{M}$ | 0.797±0.079 | 1.155±0.042 | 0.916±0.015 | 0.987±0.002 | 0.747±0.036 | 0.683±0.008 | 0.178±0.049 |
| Hard-$\mathbf{U}$ | 0.793±0.049 | 0.459±0.032 | 0.921±0.020 | 0.933±0.147 | 0.798±0.044 | 0.839±0.011 | 0.351±0.037 |
| Hard-$\mathbf{Z}$ | 0.703±0.071 | 0.480±0.021 | 0.861±0.011 | 0.920±0.122 | 0.695±0.063 | 0.668±0.007 | 0.028±0.029 |

by adding an output head to the imaging encoder; we enforce positivity of variances via a softplus operation.

Finally, we instantiate $g_\theta(\mathbf{y}, \mathbf{z})$ using the ViT imaging feature representation $\text{ViT}(\mathbf{y})$. We incorporate $\mathbf{z}$ into the mask prediction through a cross-attention mechanism. The outputs are treated as logits for the Bernoulli probabilities:

$$g_\theta(\mathbf{y}, \mathbf{z}) = \sigma\left( w^\top \left[ \text{ViT}(\mathbf{y}) \,\|\, \text{CA}\big(\mathbf{z}, \text{ViT}(\mathbf{y})\big) \right] + b \right), \tag{13}$$

where $\sigma(\cdot)$ is sigmoid, $\|$ is concatenation, and $\text{CA}(\cdot, \cdot)$ is cross-attention. Our inference procedure is shown in Fig. 3.

# 4. Synthetic Experiment

To query robustness of our inference algorithm, we generate synthetic data from the graphical model in Fig. 2 under different conditions and infer the latent variables.

## 4.1. Design and Evaluation Metrics

We train the full model and three ablations: $-\mathbf{U}$ removes the rating unreliability (forcing $\mathbf{X}_n$ to be influenced directly by $\mathbf{Y}_n$), $-\mathbf{M}$ removes the feature mask (allowing the entire $\mathbf{Y}_n$ to be explained by $\mathbf{Z}_n$), and $-\mathbf{U}, -\mathbf{M}$ removes both.

To isolate failure modes, we also evaluate the *full* model under three "noisier" generators: **Hard-M** increases the masking rate, **Hard-U** increases the scale of rating unreliability, and **Hard-Z** makes the mixture components less separable (i.e., data points are harder to distinguish apart).

We evaluate $\mathbf{Z}$ recovery via the $R_Z^2$ value after a linear alignment. We evaluate mixture recovery via clustering accuracy Acc$_C$. For rating unreliability, we report $\rho(\hat{U}, \mathbf{U})$, the mean Pearson correlation between inferred and true $\mathbf{U}$ across dimensions. For the mask $\mathbf{M}$, we evaluate both the soft and hard recovery of nuisance features through AUC$_M$ and IoU$_M$ (intersection-over-union). The latter is computed after thresholding the predicted probabilities to a

binary mask; a degenerate predictor that labels all features as nuisance attains IoU$_M$ equal to the prior that $\mathbf{M}_i = 1$. Finally, RMSE$_X$ and RMSE$_Y$ quantify the reconstruction (i.e., sampling from the latents and reconstructing the input with decoders $p_\omega, p_\phi$) errors for $\mathbf{X}$ and $\mathbf{Y}$, respectively.

## 4.2. Synthetic Data Generation

We generate $N = 2000$ observed samples $(\mathbf{X}_n, \mathbf{Y}_n)$ by first sampling mixture component $C_n \in \{1, \dots, K\}$ with $K = 4$, then draw $\mathbf{Z}_n \in \mathbb{R}^{d_Z}$ with $d_Z = 8$ from a Gaussian distribution with diagonal covariance. The auxiliary data $\mathbf{Y}_n \in \mathbb{R}^{64 \times 64}$ is a 2-D image. We sample a binary mask $\mathbf{M}_n \in \mathbb{R}^{64 \times 64}$ with pixel values generated independently: masked pixels in $\mathbf{Y}_n$ are drawn from a background Gaussian independent of $\mathbf{Z}_n$, while unmasked pixels in $\mathbf{Y}_n$ are drawn from a content-dependent Gaussian controlled by $\mathbf{Z}_n$. Finally, we draw rating unreliability $\mathbf{U}_n \in \mathbb{R}^{d_X}$ with $d_X = 12$ from a diagonal Gaussian prior whose variances depend on $\mathbf{Y}_n$, and we generate the human rating vector according to Eq. 2 with a random loading matrix $\mathbf{A}$. We provide additional experiments on data generated under mismatched ditributional assumptions in Appendix B.1.

For sensitivity analysis, Hard-$\mathbf{M}$ makes the auxiliary modality noisier by (i) increasing both the overall background feature rate to 0.5 and the variability of background features across pixels, and (ii) doubling the the background noise level, which reduces the contrast between content-driven and background pixels. Hard-$\mathbf{U}$ increases the rating ambiguity by doubling the magnitude of $\mathbf{U}_n$ and strengthening its heteroscedastic dependence on $\mathbf{Y}_n$, so that the unreliability varies more sharply across items. Hard-$\mathbf{Z}$ reduces the content identifiability by halving the distance between mixture components, while maintaining within-cluster variability.

## 4.3. Performance Analysis

Table 2 summarizes the results of our synthetic experiments. The full model simultaneously achieves strong reconstruc-

tion and high latent variable recovery. Removing $\mathbf{U}$ primarily harms the rating branch: $\mathrm{RMSE}_X$ increases substantially, while $\mathrm{RMSE}_Y$ and mask quality remain close to the full model. In contrast, removing $\mathbf{M}$ mainly harms the auxiliary data branch: $\mathrm{RMSE}_Y$ rises sharply, as forcing all pixels to be content-explained makes $\mathbf{Y}$ harder to fit. Removing both $\mathbf{U}$ and $\mathbf{M}$ affects the rating and auxiliary branches and further destabilizes the latent structure.

From the sensitivity experiments, Hard-$\mathbf{M}$ primarily degrades the recovery of $\mathbf{Y}_n$ and $\mathbf{M}_n$, resulting in a nearly double $\mathrm{RMSE}_Y$, and lower AUC/IoU. Hard-$\mathbf{U}$ primarily worsens the recovery of $\mathbf{X}$, yielding 15% higher $\mathrm{RMSE}_X$ and indicating the model uses the $\mathbf{U}$ pathway when unreliability is present. Hard-$\mathbf{Z}$ worsens the content recovery in $R_Z^2$, clustering accuracy $\mathrm{Acc}_C$, and mask inference. This is mechanistically consistent with the content-aware posterior. A more detailed analysis of performance as the level of hardness progressively increases can be found in Appendix B.2.

## 5. Real-world Experiments

### 5.1. Baseline Methods

We compare our framework against two families of baseline methods: (i) uncertainty quantification methods map the auxiliary data to ratings, and (ii) multimodal generative models learn a shared latent representation from paired data.

**Uncertainty Quantification:** Heteroscedastic Regression trains a single network to output a mean and data-dependent variance for the human ratings, yielding a predictive interval for each item (Kendall & Gal, 2017). Deep ensemble trains multiple independent regressors and uses their predictive disagreement to construct uncertainty intervals (Lakshminarayanan et al., 2017). Evidential regression predicts distributional parameters and derives the predictive intervals from the implied distribution rather than an explicit variance (Amini et al., 2020; Shi et al., 2024).

**Multimodal Models:** MVAE learns a shared latent representation with a product-of-experts fusion and uses sampling and decoding in the latent space to generate conditional predictions of human ratings from the auxiliary data (Wu & Goodman, 2018). MMVAE uses a mixture-style fusion of unimodal posteriors to learn the shared latent representation and predict the human ratings through the shared decoder (Shi et al., 2019). MoPoE trains a mixture over product-of-experts posterior defined on modality subsets and predicts the human ratings by using the subset conditioned on the auxiliary data (Sutter et al., 2021).

### 5.2. Dataset

We evaluate our method on paired neuroimaging data (resting-state fMRI) and behavioral ratings from two

*Table 3.* Summary statitics of the real-world cohorts.

| Dataset | Size | $\mathbf{X}$ Mean/Std | $\mathbf{Y}$ Mean/Std |
|---|---|---|---|
| ABIDE I | 1100 | $10.28 \pm 7.83$ | $0.36 \pm 0.25$ |
| ABIDE II | 588 | $14.14 \pm 9.74$ | $0.30 \pm 0.25$ |
| ACE | 128 | $14.08 \pm 9.68$ | $0.45 \pm 0.24$ |

datasets. ABIDE I/II is a large, multi-site, and publicly available dataset (Di Martino et al., 2014) that includes individuals diagnosed with autism spectrum disorder and neurotyical controls. Notably, the imaging acquisition procedures and behavioral phenotyping can vary across contributing sites. ACE is a complementary dataset with controlled data acquisition and a richer behavioral characterization.

Our auxiliary (objective) data consists of functional connectivity matrices computed from the resting-state fMRI (Smith et al., 2013). Following preprocessing with fMRIPrep (Esteban et al., 2019), we parcellate the brain into 116 regions and compute a $116 \times 116$ region-wise correlation matrix for each subject, which serves as $\mathbf{Y}_n$. The vector of human ratings $\mathbf{X}_n \in \mathbb{R}^{12}$ consists of twelve behavioral assessment scores, including batteries of ADI, SRS, ADOS, and SCQ (Charman & Gotham, 2013). As the phenotyping protocol differs across ABIDE sites, we omit participants with extremely sparse assessment records. Dataset summary statistics are provided in Table 3.

### 5.3. Evaluation Metrics

Given the lack of ground truth uncertainty in the real-world datasets, we evaluate the following aspects of each model: (i) ability to explain (i.e., reconstruct) the paired observations; (ii) quality of the conditional predictive distribution $p(\mathbf{x}|\mathbf{y})$. This evaluation strategy has been proposed in prior work on uncertainty modeling (Kendall & Gal, 2017).

Concretely, we report two families of metrics. For reconstruction, we measure how well the joint model explains the observed data using linear trend alignment ($R_X^2$) and standard errors ($\mathrm{RMSE}_X$, $\mathrm{RMSE}_Y$). For the conditional predictive distribution, we use a training set to learn the model parameters. Then we predict the distribution for $\mathbf{X}_n$ on held-out validation and test sets using just the auxiliary data $\mathbf{Y}_n$. Besides the coefficient of determination $R_{X,\mathrm{pred}}^2$ and standard error $\mathrm{RMSE}_{X,\mathrm{pred}}$, we also report $\mathrm{PICP}_{90}$, the empirical coverage of nominal 90% predictive intervals, and $\mathrm{MPIW}_{90}$, the mean width of these intervals. These two metrics are interpreted jointly: high coverage is only meaningful if achieved without excessively wide intervals.

### 5.4. Performance Analysis

Table 4 reports the performance of our framework and baseline methods when trained and validated on the ABIDE I/II

*Table 4.* Performance comparison on the real-world datasets. Baselines include MVAE (Multimodal VAE), MMVAE (Mixture-of-Modalities VAE), MoPoE (Mixture of Product-of-Experts), HS-Reg (Heteroscedastic Regression), Deep Ens (Deep Ensemble), and Evidential (Evidential Regression). Dash (–) indicates that the metric cannot be computed. Best performance is highlighted in bold.

| Setting | Reconstruction | | | Prediction $\mathbf{X}\vert\mathbf{Y}$ | | | |
|---|---|---|---|---|---|---|---|
| | $R^2_X \uparrow$ | $\text{RMSE}_X \downarrow$ | $\text{RMSE}_Y \downarrow$ | $R^2_{X,\text{pred}} \uparrow$ | $\text{PICP}_{90} \uparrow$ | $\text{MPIW}_{90} \downarrow$ | $\text{RMSE}_{X,\text{pred}} \downarrow$ |
| OUR METHOD | **0.976±0.005** | **2.448±0.446** | **0.149±0.038** | **0.561±0.044** | 0.891±0.035 | **10.473±0.497** | **5.932±0.489** |
| MOPOE | 0.688±0.046 | 5.016±0.469 | 0.237±0.005 | 0.210±0.085 | 0.812±0.047 | 14.822±0.450 | 7.990±0.638 |
| MMVAE | 0.609±0.025 | 5.617±0.361 | 0.235±0.005 | 0.223±0.084 | 0.830±0.025 | 12.588±0.168 | 7.914±0.460 |
| MVAE | 0.462±0.003 | 6.830±0.570 | 0.198±0.001 | 0.225±0.003 | 0.634±0.010 | 14.519±0.152 | 8.197±0.516 |
| HS-REG | – | – | – | 0.544±0.056 | 0.813±0.044 | 14.318±1.672 | 6.113±0.461 |
| DEEP ENS | – | – | – | 0.537±0.051 | 0.806±0.028 | 13.597±0.515 | 6.088±0.400 |
| EVIDENTIAL | – | – | – | 0.527±0.092 | **0.892±0.059** | 17.649±2.635 | 6.150±0.663 |
| **On Test Set** | | | | | | | |
| OUR METHOD | **0.889** | **3.322** | **0.168** | 0.501 | 0.819 | **9.262** | **7.069** |
| MOPOE | $\approx 0$ | 11.126 | 0.229 | $\approx 0$ | 0.394 | 17.404 | 14.502 |
| MMVAE | 0.289 | 8.443 | 0.223 | 0.232 | 0.723 | 21.158 | 11.558 |
| MVAE | 0.330 | 11.549 | 0.247 | 0.290 | 0.695 | 20.155 | 11.375 |
| HS-REG | – | – | – | 0.461 | 0.769 | 14.672 | 7.349 |
| DEEP ENS | – | – | – | **0.589** | **0.849** | 17.595 | 7.133 |
| EVIDENTIAL | – | – | – | 0.407 | 0.776 | 16.018 | 7.704 |

dataset. Our framework achieves near-perfect coefficient of determination $R^2_X$ and lower reconstruction errors than multimodal baselines. Specifically, the reconstruction error for the human ratings $\mathbf{X}_n$ is roughly half that of the baselines and for the neuroimaging data is 25–40% lower than the baselines. The predicted interval remains close to nominal coverage ($\text{PICP}_{90}$) and is about 20-40% tighter ($\text{MPIW}_{90}$) compared to the baseline alternatives at similar coverage, suggesting that the uncertainty is structured rather than inflated. In terms of point prediction, all models have a systematically lower performance than that of reconstruction, but our model still outperforms the baselines in terms of proportion of variance explained and errors. These trends largely persist when the models are trained on ABIDE I/II and tested on ACE, with the only exception of Deep Ensembles having slightly improved explained variance/coverage on test at a cost of nearly twice as wide intervals.

Fig. 4 visualizes the learned uncertainty across human ratings and the feature importance maps across brain networks. Panel (a) presents the mean importance mask $1 - \mathbf{M}_n$, averaged across subjects. Higher values are considered useful for uncertainty prediction. Panel (b) aggregates the importance maps across the 116 regions in our parcellation. The notable parcels cluster around midline hubs and midline cerebellum, which are implicated in the autism literature (e.g., Default mode network) (Padmanabhan et al., 2017). Panel (c) summarizes the inferred rating uncertainty, and suggests that some behavioral ratings are more reliable than others. Panel (d) provides a direct sanity check that the learned uncertainty is meaningful: when we predict ratings

from imaging alone, subjects/items with higher uncertainty variance tend to incur larger realized conditional prediction error, yielding a clear monotonic trend in the fitted line.

## 6. Conclusion

We have introduced a novel Bayesian framework to quantify item-level uncertainty in subjective human ratings without repeated measurements by leveraging an auxiliary source of data. Our graphical model disentangles a shared latent content representation from rating-specific uncertainty and simultaneously learns a a content-aware feature-selection mask over auxiliary data. Through these interactions, our framework produces data-grounded uncertainty estimates while filtering nuisance components in high-dimensional modalities. Experiments on synthetic benchmarks validate recovery and sensitivity. Our results on two neuroimaging and behavioral datasets of autism yield better reconstruction and conditional predictive distributions than state-of-the-art baseline methods in this domain. Finally, our framework produces interpretable imaging-feature patterns.

**Limitations.** Our model assumes that the data used by the human rater are either unobserved or reflect a subjective combination of data and expert judgement. While this assumption is true many real-world scenarios, it limits our ability to decompose the uncertainty by source in a more refined manner. We also note that this work would have benefited substantially from a direct comparison with empirical inter-rater uncertainty, which is a direction of future work.

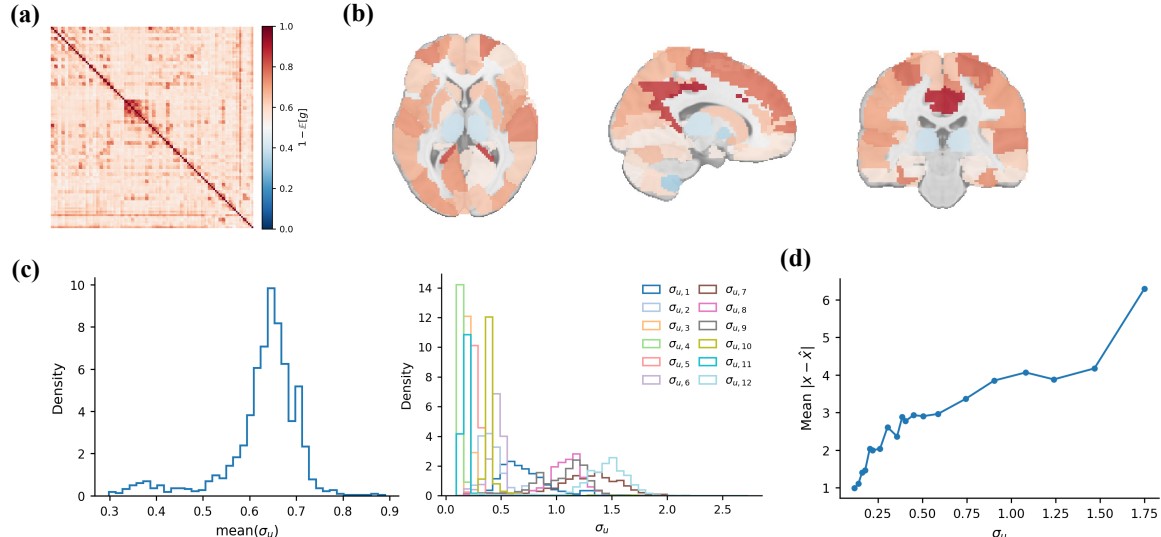

*Figure 4.* Visualization of learned useful feature probability and rating uncertainty. **(a)** Mean usefulness map over subjects **(b)** Region-level aggregation of (a) rendered on an AAL template. **(c)** Distributions of inferred rating-uncertainty standard deviations. **(d)** Alignment between predicted uncertainty and realized error

**Code Availability:** The code in this work can be downloaded at https://github.com/zijianch/BUQ-SIM

## Impact Statement

This paper presents work whose goal is to advance the field of Machine Learning. There are many potential societal consequences of our work, none which we feel must be specifically highlighted here.

## Acknowledgment

This work was supported by National Institutes of Health awards R01 HD108790 (PI Venkataraman) and R01 EB029977 (PI Caffo) and the National Science Foundation CAREER award 1845430 (PI Venkataraman).

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

## A. Derivation of ELBO and its terms

According to our assumption and the directed graph model shown in Fig. 2, we can write the generative model as a factorized joint as

$$p(\mathbf{x}, \mathbf{y}, \mathbf{z}, \mathbf{u}, \mathbf{m}, c) = p(c)p(\mathbf{z}|c)p(\mathbf{m})p(\mathbf{y}|\mathbf{z}, \mathbf{m})p(\mathbf{u}|\mathbf{y})p(\mathbf{x}|\mathbf{z}, \mathbf{u}). \tag{14}$$

Since we place an explicit Gaussian mixture structure on $\mathbf{Z}$, we have

$$p(\mathbf{z}) = \sum_{k=1}^{K} p(c=k)p(\mathbf{z} \mid c=k) = \sum_{k=1}^{K} \pi_k \mathcal{N}(\mathbf{z}; \mu_k, \sigma_k^2 I). \tag{15}$$

Subsequently, we only need to consider the joint over $(\mathbf{x}, \mathbf{y}, \mathbf{z}, \mathbf{u}, \mathbf{m})$ which is factorized as

$$p(\mathbf{x}, \mathbf{y}, \mathbf{z}, \mathbf{u}, \mathbf{m}) = p(\mathbf{z})p(\mathbf{m})p(\mathbf{y}|\mathbf{z}, \mathbf{m})p(\mathbf{u}|\mathbf{y})p(\mathbf{x}|\mathbf{z}, \mathbf{u}). \tag{16}$$

### A.1. Derivation of ELBO (Eq. 10)

We start from the marginal (log) likelihood

$$
\begin{aligned}
\log p(\mathbf{x}, \mathbf{y}) &= \log \int \sum_{\mathbf{m}} \int p(\mathbf{x}, \mathbf{y}, \mathbf{z}, \mathbf{u}, \mathbf{m}) \, \mathrm{d}\mathbf{u} \, \mathrm{d}\mathbf{z} \\
&= \log \int \sum_{\mathbf{m}} \int q(\mathbf{z}, \mathbf{u}, \mathbf{m}|\mathbf{x}, \mathbf{y}) \frac{p(\mathbf{x}, \mathbf{y}, \mathbf{z}, \mathbf{u}, \mathbf{m})}{q(\mathbf{z}, \mathbf{u}, \mathbf{m} \mid \mathbf{x}, \mathbf{y})} \, \mathrm{d}\mathbf{u} \, \mathrm{d}\mathbf{z} \\
&= \log \mathbb{E}_{q(\mathbf{z},\mathbf{u},\mathbf{m}|\mathbf{x},\mathbf{y})} \left[ \frac{p(\mathbf{x}, \mathbf{y}, \mathbf{z}, \mathbf{u}, \mathbf{m})}{q(\mathbf{z}, \mathbf{u}, \mathbf{m} \mid \mathbf{x}, \mathbf{y})} \right] \quad \geq \mathbb{E}_{q(\mathbf{z},\mathbf{u},\mathbf{m}|\mathbf{x},\mathbf{y})} \left[ \log \frac{p(\mathbf{x}, \mathbf{y}, \mathbf{z}, \mathbf{u}, \mathbf{m})}{q(\mathbf{z}, \mathbf{u}, \mathbf{m} \mid \mathbf{x}, \mathbf{y})} \right],
\end{aligned}
\tag{17}
$$

where the last step is due to Jensen's inequality. Therefore, we define the ELBO:

$$\mathcal{L}(\mathbf{x}, \mathbf{y}) = \mathbb{E}_{q(\mathbf{z},\mathbf{u},\mathbf{m}|\mathbf{x},\mathbf{y})} \left[ \log p(\mathbf{x}, \mathbf{y}, \mathbf{z}, \mathbf{u}, \mathbf{m}) - \log q(\mathbf{z}, \mathbf{u}, \mathbf{m} \mid \mathbf{x}, \mathbf{y}) \right]. \tag{18}$$

Based on our mean-field variational factorization (see Eq. 6):

$$q(\mathbf{z}, \mathbf{u}, \mathbf{m}|\mathbf{x}, \mathbf{y}) = q(\mathbf{z}|\mathbf{x}, \mathbf{y})q(\mathbf{u}|\mathbf{x}, \mathbf{y})q(\mathbf{m}|\mathbf{y}, \mathbf{z}), \tag{19}$$

we have (and we use $\mathbb{E}_q$ as a shorthand notation for $\mathbb{E}_{q(\mathbf{z},\mathbf{u},\mathbf{m}|\mathbf{x},\mathbf{y})}$)

$$
\begin{aligned}
\mathcal{L}(\mathbf{x}, \mathbf{y}) = \, &\mathbb{E}_q[\log p(\mathbf{y}|\mathbf{z}, \mathbf{m})] + \mathbb{E}_q[\log p(\mathbf{x}|\mathbf{z}, \mathbf{u})] + \mathbb{E}_q[\log p(\mathbf{z}) - \log q(\mathbf{z}|\mathbf{x}, \mathbf{y})] \\
&+ \mathbb{E}_q[\log p(\mathbf{u}|\mathbf{y}) - \log q(\mathbf{u}|\mathbf{x}, \mathbf{y})] + \mathbb{E}_q[\log p(\mathbf{m}) - \log q(\mathbf{m}|\mathbf{y}, \mathbf{z})].
\end{aligned}
\tag{20}
$$

Note that

$$
\begin{aligned}
\mathbb{E}_{q(\mathbf{z},\mathbf{u},\mathbf{m}|\mathbf{x},\mathbf{y})}[\log p(\mathbf{y}|\mathbf{z}, \mathbf{m})] &= \int \sum_{\mathbf{m}} \int q(\mathbf{z}, \mathbf{u}, \mathbf{m}|\mathbf{x}, \mathbf{y}) \log p(\mathbf{y}|\mathbf{z}, \mathbf{m}) \, \mathrm{d}\mathbf{u} \, \mathrm{d}\mathbf{z} \\
&= \int \sum_{\mathbf{m}} \left( \int q(\mathbf{u}|\mathbf{x}, \mathbf{y}) \, \mathrm{d}\mathbf{u} \right) q(\mathbf{z}|\mathbf{x}, \mathbf{y})q(\mathbf{m}|\mathbf{y}, \mathbf{z}) \log p(\mathbf{y}|\mathbf{z}, \mathbf{m}) \, \mathrm{d}\mathbf{z} \\
&= \int q(\mathbf{z}|\mathbf{x}, \mathbf{y}) \left( \mathbb{E}_{q(\mathbf{m}|\mathbf{y},\mathbf{z})}[\log p(\mathbf{y}|\mathbf{z}, \mathbf{m})] \right) \mathrm{d}\mathbf{z} \\
&= \mathbb{E}_{q(\mathbf{z}|\mathbf{x},\mathbf{y})} \left[ \mathbb{E}_{q(\mathbf{m}|\mathbf{x},\mathbf{y})}[\log p(\mathbf{y}|\mathbf{z}, \mathbf{m})] \right].
\end{aligned}
\tag{21}
$$

Similarly,

$$\mathbb{E}_{q(\mathbf{z},\mathbf{u},\mathbf{m}|\mathbf{x},\mathbf{y})}[\log p(\mathbf{x}|\mathbf{z}, \mathbf{u})] = \mathbb{E}_{q(\mathbf{z}|\mathbf{x},\mathbf{y})} \left[ \mathbb{E}_{q(\mathbf{u}|\mathbf{x},\mathbf{y})}[\log p(\mathbf{x}|\mathbf{z}, \mathbf{u})] \right]. \tag{22}$$

With the same idea, we can write

$$\mathbb{E}_{q(\mathbf{z},\mathbf{u},\mathbf{m}|\mathbf{x},\mathbf{y})}[\log p(\mathbf{z}) - \log q(\mathbf{z}|\mathbf{x}, \mathbf{y})] = \mathbb{E}_{q(\mathbf{z}|\mathbf{x},\mathbf{y})}[\log p(\mathbf{z}) - \log q(\mathbf{z}|\mathbf{x}, \mathbf{y})] = -\mathcal{D}_{\mathrm{KL}}(q(\mathbf{z}|\mathbf{x}, \mathbf{y})\|p(\mathbf{z})) \tag{23}$$

where the last equality is by the definition of KL-divergence. Similarly,

$$\mathbb{E}_{q(\mathbf{z},\mathbf{u},\mathbf{m}|\mathbf{x},\mathbf{y})}[\log p(\mathbf{u}|\mathbf{y}) - \log q(\mathbf{u}|\mathbf{x},\mathbf{y})] = \mathbb{E}_{q(\mathbf{u}|\mathbf{x},\mathbf{y})}[\log p(\mathbf{u}|\mathbf{y}) - \log q(\mathbf{u}|\mathbf{x},\mathbf{y})] = -\mathcal{D}_{\mathrm{KL}}(q(\mathbf{u}|\mathbf{x},\mathbf{y})\|p(\mathbf{u}|\mathbf{y})). \quad (24)$$

For $\mathbf{m}$, we have

$$\begin{aligned}\mathbb{E}_{q(\mathbf{z},\mathbf{u},\mathbf{m}|\mathbf{x},\mathbf{y})}[\log p(\mathbf{m}) - \log q(\mathbf{m}|\mathbf{y},\mathbf{z})] &= \mathbb{E}_{q(\mathbf{z}|\mathbf{x},\mathbf{y})}\mathbb{E}_{q(\mathbf{m}|\mathbf{y},\mathbf{z})}[\log p(\mathbf{m}) - \log q(\mathbf{m}|\mathbf{y},\mathbf{z})] \\ &= -\mathbb{E}_{q(\mathbf{m}|\mathbf{y},\mathbf{z})}\mathcal{D}_{\mathrm{KL}}(q(\mathbf{m}|\mathbf{y},\mathbf{z})\|p(\mathbf{m})).\end{aligned} \quad (25)$$

Combining these terms we obtain the desired expression Eq. 10.

### A.2. Computation of each ELBO term

First, for $\mathcal{L}_Y = \mathbb{E}_{q(\mathbf{z}|\mathbf{x},\mathbf{y})}\mathbb{E}_{q(\mathbf{m}|\mathbf{y},\mathbf{z})}[\log p_\omega(\mathbf{y}|\mathbf{z},\mathbf{m})]$, by the definition (Eq. 1), we have

$$\log p_\omega(\mathbf{y}|\mathbf{z},\mathbf{m}) = \sum_{i=1}^{d_Y} [\mathbf{m}_i \log p_{\mathrm{base}}(\mathbf{y}_i) + (1 - \mathbf{m}_i) \log p_{\mathrm{cont}}(\mathbf{y}_i \mid \mathbf{z})]. \quad (26)$$

Subsequently, by linearity,

$$\mathbb{E}_{q(\mathbf{m}|\mathbf{y},\mathbf{z})}[\log p_\omega(\mathbf{y}|\mathbf{z},\mathbf{m})] = \sum_{i=1}^{d_Y} [g_i(\mathbf{y}_i,\mathbf{z}) \log p_{\mathrm{base}}(\mathbf{y}_i) + (1 - g_i(\mathbf{y}_i,\mathbf{z})) \log p_{\mathrm{cont}}(\mathbf{y}_i \mid \mathbf{z})]. \quad (27)$$

The expectation over $\mathbb{E}_{q(\mathbf{z}|\mathbf{x},\mathbf{y})}$ cannot be computed analytically. So, we sample

$$\mathbf{z}^{(\ell)} = h_z(\mathbf{x},\mathbf{y}) + s_z(\mathbf{x},\mathbf{y}) \odot \varepsilon^{(\ell)} \quad (28)$$

for $L$ times and compute $g^{(\ell)} = g_\theta(\mathbf{y}, z^{(\ell)})$. Then,

$$\mathcal{L}_Y \approx \frac{1}{L} \sum_{\ell=1}^{L} \sum_{i=1}^{d_Y} \left[ g_i^{(\ell)} \log p_{\mathrm{base}}(\mathbf{y}_i) + (1 - g_i^{(\ell)}) \log p_{\mathrm{cont}}(\mathbf{y}_i \mid \mathbf{z}^{(\ell)}) \right]. \quad (29)$$

For $\mathcal{L}_X$, given the definition in Eq. 2, we have

$$\begin{aligned}\log p_\phi(\mathbf{x}|\mathbf{z},\mathbf{u}) &= -\frac{1}{2}\left[d_X \log(2\pi) + \log|\Sigma_X| + (\mathbf{x} - \boldsymbol{A}\mathbf{z} - \mathbf{u})^\top \Sigma_X^{-1}(\mathbf{x} - \boldsymbol{A}\mathbf{z} - \mathbf{u})\right] \\ &= -\frac{1}{2}\sum_{i=1}^{d_X}\left[\log(2\pi\sigma_{X,i}^2) + \frac{1}{\sigma_{X,i}^2}(\mathbf{x}_i - (\boldsymbol{A}\mathbf{z})_i - \mathbf{u}_i)^2\right].\end{aligned} \quad (30)$$

To compute $\mathbb{E}_{q(\mathbf{z},\mathbf{u}|\mathbf{x},\mathbf{y})}[(\mathbf{x}_i - (\boldsymbol{A}\mathbf{z})_i - \mathbf{u}_i)^2]$, we first have

$$\begin{aligned}\mathbb{E}_{q(\mathbf{z},\mathbf{u}|\mathbf{x},\mathbf{y})}[\mathbf{x}_i - (\boldsymbol{A}\mathbf{z})_i - \mathbf{u}_i] &= \mathbf{x}_i - \boldsymbol{A}_i^\top \mathbb{E}_{q(\mathbf{z}|\mathbf{x},\mathbf{y})}[\mathbf{z}] - \mathbb{E}_{q(\mathbf{u}|\mathbf{x},\mathbf{y})}[\mathbf{u}_i] \\ &= \mathbf{x}_i - \boldsymbol{A}_i^\top h_z(\mathbf{x},\mathbf{y}) - h_u(\mathbf{x},\mathbf{y})_i.\end{aligned} \quad (31)$$

Meanwhile,

$$\begin{aligned}\mathrm{Var}(\mathbf{x}_i - \boldsymbol{A}_i^\top \mathbf{z} - \mathbf{u}_i) &= \mathrm{Var}(\boldsymbol{A}_i^\top \mathbf{z}) + \mathrm{Var}(\mathbf{u}_i) \\ &= \sum_{j=1}^{d_Z} \boldsymbol{A}_{ij}^2 s_z^2(\mathbf{x},\mathbf{y})_j + s_u^2(\mathbf{x},\mathbf{y})_i\end{aligned} \quad (32)$$

Combining Eq. 31 and Eq. 32, we have

$$\mathbb{E}_{q(\mathbf{z},\mathbf{u}|\mathbf{x},\mathbf{y})}[(\mathbf{x}_i - (\boldsymbol{A}\mathbf{z})_i - \mathbf{u}_i)^2] = \left(\mathbf{x}_i - \boldsymbol{A}_i^\top h_z(\mathbf{x},\mathbf{y}) - h_u(\mathbf{x},\mathbf{y})_i\right)^2 + \sum_{j=1}^{d_Z} \boldsymbol{A}_{ij}^2 s_z^2(\mathbf{x},\mathbf{y})_j + s_u^2(\mathbf{x},\mathbf{y})_i \quad (33)$$

Subsequently,

$$\mathcal{L}_X = -\frac{1}{2} \sum_{i=1}^{d_X} \left[ \log(2\pi\sigma_{X,i}^2) + \frac{1}{\sigma_{X,i}^2} \left( \left(\mathbf{x}_i - \boldsymbol{A}_i^\top h_z(\mathbf{x}, \mathbf{y}) - h_u(\mathbf{x}, \mathbf{y})_i\right)^2 + \sum_{j=1}^{d_Z} \boldsymbol{A}_{ij}^2 s_z^2(\mathbf{x}, \mathbf{y})_j + s_u^2(\mathbf{x}, \mathbf{y})_i \right) \right]. \quad (34)$$

For the KL-divergence terms, since both $q(\mathbf{u}|\mathbf{x}, \mathbf{y})$ and $p(\mathbf{u}|\mathbf{y})$ are Gaussians (and we denote $\boldsymbol{\mu}_q = h_u(\mathbf{x}, \mathbf{y})$ and $\Sigma_q = \mathrm{diag}(s_u(\mathbf{x}, \mathbf{y})^2)$), we directly have

$$\mathcal{D}_{\mathrm{KL}}(q(\mathbf{u}|\mathbf{x}, \mathbf{y})\|p(\mathbf{u}|\mathbf{y})) = \frac{1}{2} \left[ \mathrm{tr}(\Sigma_\psi^{-1}(\mathbf{y})\Sigma_q) + \boldsymbol{\mu}_q^\top \Sigma_\psi^{-1}(\mathbf{y})\boldsymbol{\mu}_q - d_U + \log \frac{|\Sigma_\psi(\mathbf{y})|}{|\Sigma_q|} \right]$$

$$= \frac{1}{2} \sum_{i=1}^{d_U} \left[ \frac{s_{u,i}^2}{\Sigma_{\psi,ii}} + \frac{\mu_{q,i}^2}{\Sigma_{\psi,ii}} - 1 + \log \frac{\Sigma_{\psi,ii}}{s_{u,i}^2} \right]. \quad (35)$$

For the KL-divergence on the mask variable, by definition, we have

$$\mathcal{D}_{\mathrm{KL}}(q(\mathbf{m}|\mathbf{y}, \mathbf{z})\|p(\mathbf{m})) = \sum_{\mathbf{m}_i \in \{0,1\}} q(\mathbf{m}_i|\mathbf{y}_i, \mathbf{z}) \log \frac{q(\mathbf{m}_i|\mathbf{y}_i, \mathbf{z})}{p(\mathbf{m}_i)}$$

$$= \sum_{i=1}^{d_Y} \left[ g_i(\mathbf{y}_i, \mathbf{z}) \log \frac{g_i(\mathbf{y}_i, \mathbf{z})}{\rho_i} + (1 - g_i(\mathbf{y}_i, \mathbf{z})) \log \frac{1 - g_i(\mathbf{y}_i, \mathbf{z})}{1 - \rho_i} \right]. \quad (36)$$

As for the KL-divergence on the shared latent, we first have

$$\log q(\mathbf{z}|\mathbf{x}, \mathbf{y}) = -\frac{1}{2} \sum_{j=1}^{d_Z} \left[ \log(2\pi s_{z,j}(\mathbf{x}, \mathbf{y})^2) + \frac{(\mathbf{z}_j - h_{z,j}(\mathbf{x}, \mathbf{y}))^2}{s_{z,j}(\mathbf{x}, \mathbf{y})^2} \right], \quad (37)$$

and

$$\log p(\mathbf{z}) = \log \left[ \sum_{k=1}^{K} \pi_k (2\pi)^{-\frac{d_Z}{2}} \left( \prod_{j=1}^{d_Z} \left(\boldsymbol{\sigma}_{k,j}^2\right)^{-\frac{1}{2}} \right) \exp \left( -\frac{1}{2} \sum_{j=1}^{d_Z} \frac{\left(\mathbf{z}_j - \boldsymbol{\mu}_{k,j}\right)^2}{\boldsymbol{\sigma}_{k,j}^2} \right) \right]. \quad (38)$$

Note that the expectation over $q(\mathbf{z}|\mathbf{x}, \mathbf{y})$, as we have stated previously, can not be computed analytically. Therefore,

$$\mathcal{D}_{\mathrm{KL}}(q(\mathbf{z}|\mathbf{x}, \mathbf{y})\|p(\mathbf{z})) \approx \frac{1}{L} \sum_{\ell=1}^{L} \left[ \log q(\mathbf{z}^{(\ell)}|\mathbf{x}, \mathbf{y}) - \log p(\mathbf{z}^{(\ell)}) \right]. \quad (39)$$

## B. Additional Experimentation

### B.1. Model performance under mis-specified synthetic schemes

In this experiment, we query model performance under mismatched data-generation schemes. Specifically, we explore: (1) structured correlation pattern in the auxiliary modality $\mathbf{Y}$ and the mask $\mathbf{M}$ rather than the independent feature-wise corruption assumed by our model; (2) Correlated and heavy-tailed rating uncertainty $\mathbf{U}$ instead of the diagonal Gaussian covariances assumed by our model; (3) generating the shared latent content $\mathbf{Z}$ from a dual submode, non-elliptical distribution rather than Gaussian components; and (4) a combined setting where all three mis-specifications are present simultaneously.

As seen in Table 5, our model's performance has a slight degradation compared with the correct specification case, but still remains strong overall, showing that the latent decomposition remains useful under substantial model misspecification.

*Table 5.* Robustness under misspecified synthetic data-generation schemes.

| Setting | RMSE$_X$ ↓ | RMSE$_Y$ ↓ | $R_Z^2$ ↑ | Acc$_C$ ↑ | $\rho(\hat{U}, U)$ ↑ | AUC$_M$ ↑ | IoU$_M$ ↑ |
|---|---|---|---|---|---|---|---|
| Correct | 0.694±0.087 | 0.455±0.034 | 0.925±0.022 | 0.999±0.002 | 0.666±0.048 | 0.840±0.011 | 0.350±0.040 |
| Case 1 (Y) | 0.710±0.080 | 0.574±0.061 | 0.922±0.019 | 1.000±0.000 | 0.674±0.038 | 0.755±0.020 | 0.271±0.050 |
| Case 2 (U) | 0.812±0.047 | 0.459±0.038 | 0.927±0.017 | 1.000±0.000 | 0.735±0.027 | 0.838±0.013 | 0.350±0.044 |
| Case 3 (Z) | 0.651±0.064 | 0.528±0.055 | 0.904±0.017 | 0.897±0.133 | 0.592±0.114 | 0.776±0.011 | 0.255±0.114 |
| Case 4 (YUZ) | 0.780±0.050 | 0.707±0.063 | 0.869±0.018 | 0.926±0.072 | 0.704±0.072 | 0.741±0.006 | 0.229±0.042 |

## B.2. Model performance under progressive hardness increase

This experiment provides a more detailed sensitivity analysis for the full model under different level of "noisier" generators. Level 0 corresponds to the standard/base generator and level 5 corresponds to the corresponding Hard setting reported in the main text. Levels 1–4 linearly interpolate the noise severity between these two endpoints and are reported in Table 6, 7 and 8.

*Table 6.* Sensitivity to progressively stronger auxiliary-data corruption.

| LV | RMSE$_X$ ↓ | RMSE$_Y$ ↓ | $R^2_Z$ ↑ | Acc$_C$ ↑ | $\rho(\hat{U}, U)$ ↑ | AUC$_M$ ↑ | IoU$_M$ ↑ |
|---|---|---|---|---|---|---|---|
| 1 | $0.703 \pm 0.085$ | $0.458 \pm 0.038$ | $0.926 \pm 0.021$ | $1.000 \pm 0.001$ | $0.647 \pm 0.050$ | $0.840 \pm 0.012$ | $0.350 \pm 0.041$ |
| 2 | $0.724 \pm 0.085$ | $0.570 \pm 0.047$ | $0.929 \pm 0.011$ | $0.999 \pm 0.001$ | $0.691 \pm 0.048$ | $0.798 \pm 0.019$ | $0.302 \pm 0.055$ |
| 3 | $0.739 \pm 0.083$ | $0.726 \pm 0.054$ | $0.926 \pm 0.014$ | $0.998 \pm 0.002$ | $0.716 \pm 0.049$ | $0.759 \pm 0.023$ | $0.253 \pm 0.058$ |
| 4 | $0.761 \pm 0.064$ | $0.924 \pm 0.054$ | $0.920 \pm 0.011$ | $0.939 \pm 0.135$ | $0.731 \pm 0.039$ | $0.722 \pm 0.020$ | $0.211 \pm 0.063$ |

*Table 7.* Sensitivity to progressively stronger rating uncertainty.

| LV | RMSE$_X$ ↓ | RMSE$_Y$ ↓ | $R^2_Z$ ↑ | Acc$_C$ ↑ | $\rho(\hat{U}, U)$ ↑ | AUC$_M$ ↑ | IoU$_M$ ↑ |
|---|---|---|---|---|---|---|---|
| 1 | $0.713 \pm 0.091$ | $0.461 \pm 0.033$ | $0.919 \pm 0.018$ | $0.997 \pm 0.002$ | $0.649 \pm 0.070$ | $0.840 \pm 0.012$ | $0.348 \pm 0.039$ |
| 2 | $0.756 \pm 0.086$ | $0.463 \pm 0.035$ | $0.924 \pm 0.015$ | $0.999 \pm 0.001$ | $0.709 \pm 0.044$ | $0.840 \pm 0.011$ | $0.351 \pm 0.036$ |
| 3 | $0.772 \pm 0.063$ | $0.458 \pm 0.035$ | $0.927 \pm 0.017$ | $0.998 \pm 0.002$ | $0.747 \pm 0.046$ | $0.839 \pm 0.013$ | $0.349 \pm 0.041$ |
| 4 | $0.789 \pm 0.043$ | $0.458 \pm 0.030$ | $0.921 \pm 0.013$ | $1.000 \pm 0.001$ | $0.767 \pm 0.047$ | $0.839 \pm 0.012$ | $0.349 \pm 0.035$ |

*Table 8.* Sensitivity to progressively reduced latent-cluster separation.

| LV | RMSE$_X$ ↓ | RMSE$_Y$ ↓ | $R^2_Z$ ↑ | Acc$_C$ ↑ | $\rho(\hat{U}, U)$ ↑ | AUC$_M$ ↑ | IoU$_M$ ↑ |
|---|---|---|---|---|---|---|---|
| 1 | $0.715 \pm 0.084$ | $0.459 \pm 0.034$ | $0.922 \pm 0.013$ | $0.998 \pm 0.002$ | $0.661 \pm 0.052$ | $0.841 \pm 0.012$ | $0.352 \pm 0.042$ |
| 2 | $0.707 \pm 0.082$ | $0.451 \pm 0.031$ | $0.921 \pm 0.013$ | $1.000 \pm 0.001$ | $0.687 \pm 0.047$ | $0.814 \pm 0.016$ | $0.303 \pm 0.046$ |
| 3 | $0.711 \pm 0.074$ | $0.458 \pm 0.036$ | $0.904 \pm 0.016$ | $0.995 \pm 0.006$ | $0.676 \pm 0.076$ | $0.771 \pm 0.015$ | $0.221 \pm 0.050$ |
| 4 | $0.716 \pm 0.077$ | $0.477 \pm 0.041$ | $0.887 \pm 0.033$ | $0.993 \pm 0.006$ | $0.699 \pm 0.065$ | $0.723 \pm 0.017$ | $0.116 \pm 0.058$ |

