# OpenReview forum: "A Bayesian Approach to Quantify the Uncertainty of Human Ratings in a Single-Instance Multimodal Framework"
_ICML.cc/2026/Conference — ICML 2026 regular_

### Official Review · Reviewer_MShC · 2026-03-10

**Soundness:** 3
**Presentation:** 3
**Significance:** 2
**Originality:** 2
**Overall Recommendation:** 3
**Confidence:** 4

**Summary:**

This paper introduces a Bayesian approach designed to quantify uncertainty in subjective human ratings by leveraging auxiliary objective data. The framework utilizes a shared latent representation to capture common factors between ratings and objective inputs, while a data-conditioned prior accounts for fluctuations in human judgment. In addition to uncertainty, the model predicts a mask over the objective data to identify specific features that are effectively captured or explained by the shared latent vector. To ensure scalability, the authors implement an amortized variational inference procedure using neural encoders and decoders. Validation through synthetic stress tests and real-world fMRI-based autism assessments demonstrates the model's ability to recover latent uncertainty when repeated measurements are unavailable.

**Compliance With Llm Reviewing Policy:**

Affirmed.

**Final Justification:**

This paper presents a well-structured Bayesian framework for quantifying uncertainty in subjective human ratings and demonstrates notable strengths, particularly in its clear mathematical derivations. However, critical limitations including the omission of the original source data within the graphical model and the absence of a baseline comparison against empirical multi-rater uncertainty significantly undermine the evaluation. While the underlying concept is promising, these shortcomings make me lean to reject the submission.

**Key Questions For Authors:**

1. In the synthetic experiments, the observed reconstruction errors are higher than one would typically anticipate for a controlled setting. Could the authors explain this?

2. To further validate the model’s sensitivity, could the authors demonstrate a controlled experiment where the input data (e.g., images) is subjected to incremental levels of noise or artifacts? It would be highly compelling to see if the predicted uncertainty correlates linearly with these known levels of data degradation.

3. High uncertainty typically implies a broad distribution of possible human labels. Can the proposed method be adapted to suggest plausible alternative ratings?

4. Could the authors provide a theoretical justification for modeling uncertainty as an additive component to the mean rather than a scaling factor for the variance? In traditional Bayesian frameworks, uncertainty is usually captured by the spread ($\sigma^2$) of the distribution. Please clarify why an additive mean shift is a more appropriate representation in this specific architecture.

**Limitations:**

The paper lacks a dedicated section addressing its limitations, similar to the concerns noted above.

**Strengths And Weaknesses:**

Strengths
1. The manuscript is well-structured and written with high clarity, providing all necessary context for the reader.

2. Both the problem formulation and the mathematical derivation of the proposed solutions are rigorous and detailed. The proposed solution is well-justified, and experiments on both synthetic and real-world data yield promising results.

Weaknesses
1. The Bayesian graphical model seems to overlook a critical observed variable: the specific data upon which the human ratings were based. For instance, if ratings are based on low-resolution ultrasound images while the objective data is high-resolution MRI, the uncertainty in the ratings may stem from the limitations of the ultrasound itself. Formulating uncertainty as a random variable conditioned only on objective data may not fully capture these discrepancies.

2. The derivations rely on several assumptions regarding underlying distributions. These choices are currently under-explained. The authors should provide stronger motivation for these selections or include ablations comparing different distributional choices.

3. Since the authors note that multiple ratings are the standard (though expensive) way to assess uncertainty, the paper would be significantly strengthened by an experiment comparing the model’s predicted uncertainty (from a single item) against the empirical uncertainty derived from a multi-rater dataset. This would directly demonstrate the model's ability to reduce the need for redundant manual labor.

4. The manuscript would benefit from a high-level conceptual overview of the method before transitioning into dense technical and distributional assumptions.



Minor Issues
1. In real-world scenarios, features of objective data are rarely independent. Therefore, the assumption in Equation (1) may be overly restrictive.

2. Regarding Equation (2), it seems more theoretically sound for uncertainty to affect the variance of the distribution rather than acting as an additive contribution to the mean. This specific design choice requires further clarification.

3. In both synthetic and real-world experiments, the authors lack visual results demonstrating that the computed uncertainty or masks align with identifiable regions of interest in the objective data.

---

> ### Author Rebuttal · Authors · 2026-03-30
>
> We thank the reviewer for their constructive comments. Below, we address each concern and hope to clarify any ambiguities.
>
> (W1) In many applications, the data used by the human rater is not observed, and might involve multiple data sources and “expert judgement”. By contrast, the auxiliary data modality (e.g., imaging) is objective. From a modeling standpoint, the random variable $U$ does not estimate uncertainty from the auxiliary modality alone, but is also grounded in the human ratings. This can be seen through the posterior $p(u \mid x, y)$ and its variational approximation $q(u \mid x, y)$. The objective-conditioned term $p(u \mid y)$ is *only* the prior used to regularize the latent variable $U$ and is not the model output.
>
> (W2) Our model contains the following main variables: human rating $X$ (continuous), auxiliary data $Y$ (continuous in our case), shared latent content $Z$ (continuous), uncertainty $U$ (continuous), and feature selection mask $M$ (binary). Distributional choices are made based on the nature of these variables. For example, a (mixture) Gaussian distribution (1) can represent continuous random variables ($X$, $Y$, $Z$, $U$), (2) is widely adopted in VAEs and generative models, and (3) ensures tractability in the learning. Likewise, a Bernoulli mask $M$ is often used to model feature selection. While the forward model distributions are selected for simplicity interpretability, the inference is parametrized by deep neural networks, which allows the model to capture nonlinear cross-modal relationships.
>
> (W3) We agree that comparison against empirical inter-rater uncertainty would be the strongest direct evaluation. However, most real-world datasets, especially in the medical domain, do not include repeated ratings, which is precisely the regime that motivates the method. Instead, our validation uses (i) direct recovery on synthetic data with known latent uncertainty and (ii) indirect real-data validation via predictive calibration/sharpness and error alignment. We will state this limitation more explicitly.
>
> (W4) We agree that high-level intuition should be provided upfront. We will augment the existing model overview in Fig. 1 and Section 3.1 with a short conceptual paragraph.
>
> (M1) Our factorization in Eq. (1) is conditioned on both $Z$ and $M$. Specifically, $Z$ captures the shared information between the ratings and objective data, and $M$ indicates which feature is relevant and which is not. Given these pieces of information, the features can be decorrelated. We agree that the objective features are NOT *marginally independent*, and our framework supports this point.
>
> (M2 **& Q4**) Although Eq. (2) is written as an additive latent residual, the marginalization of $U$ induces a variance-based uncertainty, as intuited by the reviewer. Specifically, after marginalization, the uncertainty contributes implicitly to the heteroscedasticity of the human rating ($ X \mid Z, Y \sim \mathcal{N}(Az, \operatorname{diag}(\sigma_X^2)+\Sigma_\psi(y)) $) through the input-dependent term $\Sigma_\psi(y)$. We use an additive form because it provides an interpretable decomposition between the shared information that controls objective data as well as the human ratings ($Az$), and the rating-specific uncertainty ($U$). This strategy also enables sampling of plausible alternative rating realizations rather than only outputting a variance scalar.
>
> (M3) We may not have highlighted this clearly enough: Fig. 4(a-c) already visualizes the learned feature selection mask and inferred uncertainty distributions, and Fig. 4(d) links uncertainty to realized rating prediction error. We will make these visual results more explicit in the text.
>
> (Q1) The synthetic experiment is controlled but is not deterministic. By construction, both modalities contain irreducible stochasticity. The reported RMSE is also computed on sampled reconstructions from the generative model rather than on noise-free conditional means. Therefore, a near-zero RMSE is not expected even under correct specification. With that said, our RMSE is still much smaller than the generated sample variance ($X$: mean=0.72, var=2.25; $Y$: mean=0.01, var=1.85 for the base case).
>
> (Q2) We have run new synthetic experiments that interpolate from the baseline generator to the corresponding hard setting in the paper for the image, uncertainty and shared latent. The results suggest that (1) the increasing difficulty mainly affects the corresponding (rating or objective) branch; and (2) performance degrades as we increase difficulty. The full performance table can be found at
>
> https://anonymous.4open.science/r/anonymous_table-7DC6/table_incrementalArtifacts.pdf
>
> (Q3) Yes. Because the framework is generative, it can produce samples from the conditional rating distribution rather than only point predictions or interval widths.
>
> (L1) We will add an explicit paragraph on the limitations of our method based on the points noted above by the reviewer.

---

> > ### Author Rebuttal · Reviewer_MShC · 2026-04-03
> >
> > I appreciate the detailed rebuttal and I have no additional questions.

---

> > > ### Author Response · Authors · 2026-04-06
> > >
> > > We again thank the reviewer for the constructive comments, and for noting that the concerns have been fully resolved.

---

### Official Review · Reviewer_PFof · 2026-03-13

**Soundness:** 4
**Presentation:** 4
**Significance:** 4
**Originality:** 4
**Overall Recommendation:** 6
**Confidence:** 3

**Summary:**

This paper proposes a Bayesian multimodal framework for estimating item-level uncertainty in subjective human ratings without repeated annotations, by leveraging paired auxiliary objective data. The model introduces a shared latent content variable, a rating-specific uncertainty latent variable, and a feature-selection mask over the auxiliary modality, and trains these components with amortized variational inference. The empirical study includes synthetic recovery/ablation experiments and real-world neuroimaging/behavioral experiments, where the method improves over several uncertainty-aware and multimodal baselines on reconstruction and conditional prediction metrics.

**Compliance With Llm Reviewing Policy:**

Affirmed.

**Final Justification:**

The rebuttal fully addressed my concerns. I maintain the Strong Accept recommendation accordingly.

**Key Questions For Authors:**

1. What practical/statistical challenges do you foresee in transferring this framework to auxiliary modalities beyond neuroimaging?
2. In the absence of repeated ratings, what evidence most strongly supports interpreting $U$ as human-rating unreliability rather than generic unpredictability?

**Limitations:**

Yes

**Strengths And Weaknesses:**

Strengths: The problem is important and practically relevant; the latent-variable decomposition is conceptually appealing; the manuscript is exceptionally well written and easy to follow; and the empirical study includes both controlled synthetic tests and real-world experiments with competitive baselines.
Weaknesses (Minor): The real-world setting lacks direct ground truth for rating uncertainty, so the validation of the most central latent quantity is indirect. But this is understandable given the problem setting, and the real-world problems still highlight the value of the proposed method; the authors may offer additional insights into the practical challenges in transferring the framework beyond neuroimaging and behavioral assessment to other auxiliary modalities.

---

> ### Author Rebuttal · Authors · 2026-03-30
>
> We thank the reviewer for their appreciation of our work and constructive feedback.
>
> **(W1 & Q1)** We value the opportunity to discuss the possible broader applications of our framework. Neuroimaging is just one instantiation rather than a specific requirement of the model. Another immediate example can be ECGs (time series), medical records (texts, lab results). In these cases, the main transfer challenges would be (i) ensuring that the auxiliary modality aligns enough with the subjective rating, and (ii) choosing a proper likelihood distribution that is matched to the auxiliary data type and that handles the possible structured dependencies.
>
> (W2) We acknowledge the lack of ground truth in this type of study, and we appreciate the reviewer for understanding such a scenario. To demonstrate the value of our framework in the absence of ground-truth, we followed the strategies adopted in previous uncertainty quantification work. These include assessing conditional predictive distribution and using PICP90/MPIW90 to evaluate stochasticity. Our results also show a monotonic trend between predicted uncertainty and realized error, meaning that if the model deem that a subject/item is more uncertain, that subject/item is indeed harder to be reliably predicted.
>
> (Q2) Structurally, $U$ is the rating-specific random effect that remains after accounting for the shared underlying content (such as biology states in our case) between human ratings and the auxiliary data (i.e., neuroimaging data in our setting).  Moreover, after marginalization, $U$ contributes exactly as heteroscedastic variance of the rating ($\mathcal{N}(Az, \operatorname{diag}(\sigma_X^2)+\Sigma_\psi(y))$ ) instead of an additive component into the rating. Empirically, we showed in Table 2 that removing $U$ primarily hurts the $X$ branch, which again justifies that $U$ captures the uncertainty specific to ratings rather than generic unpredictability.

---

> > ### Author Rebuttal · Reviewer_PFof · 2026-04-01
> >
> > The authors have provided thorough clarifications that adequately address my initial concerns. After reviewing the other official reviews and the corresponding rebuttals, I have decided to maintain my original rating. The work represents a clear advancement in the literature, and I find no compelling reason to delay its publication.

---

> > > ### Author Response · Authors · 2026-04-06
> > >
> > > We greatly appreciate the reviewer for acknowledging our response, and for appreciating our work.

---

### Official Review · Reviewer_Fn97 · 2026-03-16

**Soundness:** 2
**Presentation:** 3
**Significance:** 3
**Originality:** 2
**Overall Recommendation:** 5
**Confidence:** 4

**Summary:**

This paper studies an important problem: estimating item-level uncertainty in subjective human ratings from a single observation paired with auxiliary objective data. The proposed Bayesian framework disentangles shared content, rating uncertainty, and a feature-selection mask over the auxiliary modality, and is trained with amortized variational inference.

**Compliance With Llm Reviewing Policy:**

Affirmed.

**Final Justification:**

The authors have addressed all of my concerns and provided many additional experiments. These updates have strengthened the paper, and I now lean toward recommending acceptance.

**Key Questions For Authors:**

Q1. Can you provide stronger indirect validation that the inferred uncertainty is meaningful on real data? For example, does higher estimated uncertainty correlate with stronger cross-site variability or larger disagreement-related instability?
Q2. How many trainable parameters does the full model have for each real-data experiment?

**Strengths And Weaknesses:**

S1. The problem setting is meaningful and practically relevant, especially in domains where repeated annotations are expensive or unavailable.

S2. The probabilistic formulation is reasonably novel in how it combines shared latent content, uncertainty modeling, and feature masking in one framework.

S3. The synthetic experiments and ablations are organized and suggest that different components of the model play distinct roles.

W1. The main contribution is uncertainty quantification, but on the real datasets there is no ground-truth uncertainty. As a result, the evaluation reduces to reconstruction and conditional prediction quality, which only provide indirect evidence. This makes the strongest claim of the paper insufficiently supported.

W2. Since the synthetic data are generated from the same model family, the results mainly show recoverability under correct specification, rather than robustness under realistic mismatch. This limits how much the synthetic results can support the real-world claim.

---

> ### Author Rebuttal · Authors · 2026-03-30
>
> We thank the reviewer for their constructive comments.
>
> (W1) We acknowledge the lack of ground truth in the real-world datasets. This motivates our use of synthetic experiments, for which the ground-truth uncertainties are available. Our synthetic experiments are designed to query robustness in the following ways: (1) From the observed data pair (ratings, objective data) how well does the model recover the ground-truth “shared underlying content” ($Z$) that generates the given data pair and the uncertainty in the ratings? (2)  Are the effects of the shared underlying content $Z$ and the uncertainty $U$ distinct? (3) How does the performance change if we change the way of data generation? In the real-world dataset, to demonstrate the value of our framework in the absence of ground-truth, we followed the strategies adopted in previous uncertainty quantification work. These include assessing conditional predictive distribution and using PICP90/MPIW90 to evaluate stochasticity. Our results also show a monotonic trend between predicted uncertainty and realized error, implying that samples which the model deems to be more uncertain are indeed harder to predict.
>
> (W2) We thank the reviewer for noting this issue. We have run four additional experiments to query model performance under mismatched data-generation schemes. Specifically, we explore: (1) structured correlation pattern in the auxiliary modality $Y$ and the mask $M$ rather than the independent feature-wise corruption assumed by our model; (2) Correlated and heavy-tailed rating uncertainty $U$ instead of the diagonal Gaussian covariances assumed by our model; (3) generating the shared latent content $Z$ from a dual submode, non-elliptical distribution rather than Gaussian components; and (4) a combined setting where all three misspecifications are present simultaneously. Our model’s performance has a slight degradation compared with the correct specification case, but still remains strong overall, showing that the latent decomposition remains useful under substantial model misspecification. The full performance table can be found at
>
> https://anonymous.4open.science/r/anonymous_table-7DC6/table_misspecification.pdf
>
> (Q1) We thank the reviewer for this constructive question. We first note that our real-world dataset does not include repeated human ratings to directly quantify disagreement. That said, we have provided an “uncertainty vs. rating prediction deviation” plot (Fig. 4d of the manuscript), which suggests that a higher uncertainty is related to a larger rating prediction deviation. This result can be viewed as an indirect form of large disagreement-related instability. To explore cross-site variability, we have conducted an additional analysis, in which we compute pairwise site differences of the modalities and define a “site shift” as the mean difference for all other sites. We then correlate the site-level shift with the average inferred uncertainty at that site. This analysis yields a Spearman correlation in ABIDE1 of 0.32 and in ABIDE2 of 0.84, implying that sites which are more different from other sites have higher average uncertainty.
>
> (Q2) The full model has around 0.8M parameters to learn, of which around 0.45M are for image decoders and 0.15M for ViT encoders. We note that this number is already comparatively smaller than traditional VAE/GAN structure.

---

> > ### Author Rebuttal · Reviewer_Fn97 · 2026-04-04
> >
> > Thank you for the thorough rebuttal. I find the motivation and setting of this work similar to those in a recent paper I read on human-centered assessment, which also includes subject-objective collaboration (https://openreview.net/forum?id=XNbVoi9mfr). The specific topics may differ, but they are still worth including in the discussion. Moreover, that paper designs real-world experiments by introducing noise to existing collected data, which may inspire you to design your own experiments.

---

> > > ### Author Response · Authors · 2026-04-06
> > >
> > > We thank the reviewer for acknowledging our response and for bringing this paper to our attention. The referenced paper was published on 01/26/2026, just as our manuscript entered review for ICML, and therefore represents concurrent, independently developed work. The main difference between our work and the referenced paper is that the latter aims to calibrate scores derived from the objective source (i.e., model output) by projecting it onto human comparative judgement. In contrast, our goal is to use objective data to separate true signals from uncertainty in noisy human ratings. We will include a brief discussion in the manuscript.
> > >
> > > With that said, the reviewer’s suggestion to inject noise into the real-world objective data, as done in the referenced paper, is an interesting experiment. To query this behavior, we kept the human ratings fixed and degraded the image side by injecting noise (1) at levels that are comparative to original data standard deviation, and (2) randomly across the image.
> > >
> > > | Setting | $R^2_{X, \text{pred}}$ | $\mathrm{PICP}_{90}$ | $\mathrm{MPIW}_{90}$ | $\mathrm{RMSE}_{X,\text{pred}}$ | $\sigma_U$         | $\text{Spearman}(\sigma_U, \|x-\hat{x}\|)$ |
> > > |---------|---------------------------|----------------------|----------------------|---------------------------------|--------------------|------------------------------|
> > > | Base Case     | $0.561\pm0.044$           | $0.891\pm0.035$      | $10.473\pm0.497$     | $5.932\pm0.489$                 | $ 0.604 \pm 0.130$ | $ 0.585 \pm 0.077$           |
> > > | 0.5     | $0.490\pm 0.088$          | $0.883\pm 0.045$     | $10.549 \pm 0.627$   | $6.384\pm0.608$                 | $ 0.616 \pm 0.133$ | $ 0.597 \pm 0.081$           |
> > > | 1.0     | $0.460\pm 0.098$          | $0.878\pm 0.046$     | $11.321 \pm 0.618$   | $6.559\pm 0.647$                | $ 0.625 \pm 0.132$ | $ 0.563 \pm 0.063$           |
> > >
> > > The result shows that for the prediction part (given image $Y$ predict human rating $X$): (1) the prediction of $X$ degrades for both point and interval estimation since $Y$ is now the only evidence to predict $X$; and (2) the uncertainty estimation stays roughly flat (0.604 vs. 0.616 vs. 0.625).  Under our formulation, where uncertainty is modeled as a latent variable capturing intrinsic noise in human ratings, this behavior is consistent with the intended separation between input-driven prediction error and item-specific uncertainty.

---

### Decision · Program_Chairs · 2026-04-30

**Decision:**

Accept (regular)

**Comment:**

The paper proposes a Bayesian approach to estimate item-level uncertainty in subjective human ratings without requiring repeated annotations, by leveraging paired auxiliary objective data. It employs a latent representation to disentangle shared content from fluctuations in human judgment. Experiments on both synthetic and real data demonstrate the effectiveness of the proposed method compared to baseline approaches.  Reviewers highlight several strengths, including a novel probabilistic formulation, the importance of the practical problem addressed, convincing experimental results, and a well-written presentation. They also identify several weaknesses, such as the lack of modeling for specific data characteristics within the Bayesian framework, the absence of direct uncertainty quantification (UQ) evaluation on real data, missing comparisons between the estimated uncertainty and empirical uncertainty derived from multiple annotators, and the need for further validation across additional data modalities. The authors’ rebuttal addresses most of the concerns; however, some issues remain, particularly the lack of modeling of specific data characteristics and the missing comparison between estimated and empirical uncertainty from multiple users. The final ratings for the paper are one strong accept, one accept, and one weak reject.